# Low-bit Quantization for Seeing in the Dark

## Abstract

Several properties of raw data exhibit significant potential for enhancing images under extremely low-light conditions. Recently, many deep-learning methods for raw-based low-light image enhancement (LLIE) have demonstrated excellent performance. However, deploying them on resource-limited devices is restricted by high computational and storage demands. In this work, we propose a novel low-bit quantization method for raw-based LLIE model to improve their efficiency. Nevertheless, directly adopting existing quantizers for LLIE networks leads to obvious performance drop due to two main reasons. *i)* The U-Net model, commonly employed in LLIE, faces challenges in identifying a suitable quantization range due to disparities in distribution between the encoder and decoder features. *ii)* Low-bit quantized LLIE networks struggle to restore clear details in low-light images because their features have a constraint capacity. We address these issues by introducing a novel low-bit quantization method, the Distribution-Separative Asymmetric Quantizer (DSAQ), designed specifically for U-Net architectures used in LLIE. In order to accurately determine the quantization intervals, DSAQ separates the distribution of encoder and decoder features before they are concatenated by the skip connection. We also make the quantizer asymmetric with trainable scale and offset parameters to suit skewed activation ranges caused by non-linear functions. To further enhance performance, we propose a uniform feature distillation technique, which allows the low-bit student model to effectively assimilate knowledge from the full-precision teacher model, bridging the gap in representation capability. Extensive experiments show that our approach not only greatly reduces the memory and computational requirements of raw-based LLIE models but also has a promising performance. Our low-bit quantized model can achieve comparable or superior results to full-precision counterparts.

## 1 Introduction

Capturing high-quality images under extremely low-light conditions is important for night surveillance and various downstream computer vision tasks Hong et al. (2021); Chen et al. (2023). However, the degradations including low signal-to-noise ratio (SNR) and obvious color cast caused by limited photon count make it challenging. Low-light image enhancement (LLIE) methods provide a post-capture solution that prevents noise amplification at high ISO settings and motion blur due to long exposure time.

Recently, deep-learning LLIE methods trained with paired datasets Chen et al. (2018); Dong et al. (2022); Wei et al. (2018) have shown outstanding performance. In this work, we focus on enhancing low-light raw images because of their inherent advantages for LLIE Wei et al. (2022); Huang et al. (2022). On the one hand, they maintain a linear relationship with photon counts and a tractable noise distribution before passing through the image signal processing (ISP) pipeline. On the other hand, they have a higher bit-depth that can distinguish subtle low-intensity details. Although deep-learning models for raw-based LLIE Chen et al. (2018); Zhu et al. (2020); Jin et al. (2023) have achieved promising results, the deployment of these neural networks on edge devices like mobile phones or embedded cameras is hindered by their high computational and storage demands.

A potential solution to this problem lies in the technique of network quantization Zhou et al. (2016); Li et al. (2020); Qin et al. (2023). Quantization involves converting the continuous weights and activations (features) of a neural network into discrete low-bit representations, significantly reducing the model's memory footprint and accelerating its inference speed. In this work, we intend to quantize

the LLIE model to a range of 2-4 bits to achieve a higher compression ratio. Despite its benefits in terms of efficiency, low-bit network quantization may lead to a deterioration in model performance.

For raw-based LLIE methods, we recognize two main reasons for this performance degradation. **Firstly**, many of raw LLIE networks are based on U-Net Ronneberger et al. (2015) structure Chen et al. (2018); Dong et al. (2022); Huang et al. (2022). In the U-Net architecture, we observe that the features from the encoder and decoder, concatenated through skip connections, show notable differences in distribution. Moreover, the use of non-linear activation functions, such as LeakyReLU Chen et al. (2018); Lamba & Mitra (2021), results in asymmetric distributions of positive and negative values. These factors pose challenges in accurately determining the quantization range. **Secondly**, features in low-bit quantized networks exhibit a representation capability gap compared to those in full-precision networks. Therefore, existing knowledge distillation schemes Li et al. (2020); Zhong et al. (2022), which directly impose constraints on normalized features, cannot fully transfer intrinsic semantic information from full-precision teacher model to the low-bit quantized student model.

In this paper, we present a novel low-bit quantization method for raw-based LLIE to solve the above problems. Specifically, we propose a Distribution-Separative Asymmetric Quantizer (DSAQ) that is tailored for U-Net based LLIE method. It quantizes the encoder and decoder features respectively before concatenation to facilitate the learning of quantization interval. In order to mitigate the influence of non-linear functions on the distribution of activations, we introduce trainable scale and offset parameters to implement the asymmetric quantizer. We further introduce a uniform feature distillation that maps features of quantized student model and full-precision teacher model into a uniform latent feature space. So low-bit network can better obtain intrinsic information from teacher model and restore clearer details from low-light images. Through extensive experiments, we demonstrate that our quantization method surpasses previous quantizers in raw LLIE and the low-bit network achieves comparable enhancement results to their full-precision counterparts. Our main contributions can be summarized as follows:

- We propose a compact low-bit quantized model for low-light raw image enhancement, which can achieve satisfactory results with low memory and computation.
- We build a Distribution-Separative Asymmetric Quantizer (DSAQ) for U-Net structure. It separately determines the quantizer of different features before concatenation and introduces asymmetric quantization for activations with skewed distribution.
- We design a uniformed feature distillation that reduces capacity difference between features in quantized and full-precision models. So the knowledge from teacher model can be easily transferred to student model.

## 2 RELATED WORK

In this section, we first review deep-learning methods for raw-based LLIE. We then review some quantization techniques for efficient neural network inference.

### 2.1 RAW-BASED LOW-LIGHT IMAGE ENHANCEMENT

Because of the merits of raw images discussed in Section 1, they are commonly used for LLIE in extremely dark environments. The pioneering work Chen et al. (2018) builds a large-scale paired short/long exposure raw image dataset for LLIE, dubbed See-in-the-Dark (SID). A U-Net is employed for restoring noisy low-light raw input into bright RGB images. A parallel work DeepISP Schwartz et al. (2019) uses an end-to-end neural network to process low-light raw images, which achieves better visual quality than manufactured ISP. Based on the SID dataset, following work also introduces residual learning Maharjan et al. (2019), self-guidance strategy Gu et al. (2019) and multi-criterion loss Zamir et al. (2021) for single-stage raw to RGB LLIE.

Another line of methods decompose the problem of raw-based LLIE into different aspects and design multi-stage networks. EEMEFN Zhu et al. (2020) sequentially performs multi-exposure fusion and edge enhancement for LLIE. LDC Xu et al. (2020) enhances the low-frequency part and reconstructs the high-frequency details of the low-light images in two consecutive stages. MCR Dong et al. (2022) first learns to synthesize monochrome images with additional supervision. Then a dual-branch network is leveraged to fuse generated monochrome and color images to produce enhanced RGB

results. Huang *et al.* Huang et al. (2022) proposes a raw-guiding exposure enhancement network, which consist of three cascaded U-Nets for unprocessing, denoising and processing. DNF Jin et al. (2023) decouples raw-based LLIE into raw image denoising stage and RGB image color correction stage to mitigate the domain ambiguity. A feedback module enables feature interaction across two stages to reduce error accumulation.

The power of these neural networks to see in the dark relies on their model depth and computational complexity. Some work Lamba & Mitra (2021); Lamba et al. (2020) also improves efficiency of LLIE models by designing lightweight network architectures. In this work, we resort to network quantization to achieve efficient LLIE.

## 2.2 Neural Network Quantization

Neural network quantization involves reducing the precision of weights and activations in a neural network, representing them with a lower-bit (usually 2-8 bits) discrete representation Nagel et al. (2021). This process can effectively reduce the model size and computation cost, and it can be incorporated with other network compression techniques like parameter pruning Zhang et al. (2022); Wang & Fu (2023) and knowledge distillation Zhu et al. (2023); Li et al. (2020). There are two primary paradigms for network quantization: Post-Training Quantization (PTQ) Hubara et al. (2021); Li et al. (2021) and Quantization Aware Training (QAT) Zhou et al. (2016); Choi et al. (2018); Li et al. (2022; 2023); Esser et al. (2020). PTQ methods allow for the efficient quantization of pre-trained neural networks with minimal data and no retraining. However, they suffer from sub-optimal performance due to fixed parameters and limited fine-tuning capabilities. In this paper, we adopt QAT that retrain the network parameters with simulated quantization and full training data to achieve a better performance in low-bit (*i.e.*, less than 4 bits) quantization. Additionally, 1-bit quantization methods (also known as binary neural networks) are not discussed because they often rely on specific designs to avoid severe performance degradation Liu et al. (2018); Cai et al. (2023) and have different hardware implementations Qin et al. (2023).

Low-bit network quantization with QAT has been widely applied to various computer vision tasks, with early efforts primarily focusing on the quantization of classification models Choi et al. (2018); Jacob et al. (2018); Gong et al. (2019); Jung et al. (2019); Esser et al. (2020); Bhalgat et al. (2020). These methods incorporate either a learnable quantization interval Choi et al. (2018) or a learnable step size Esser et al. (2020) within the quantizer, optimizing these parameters along with network weights to minimize task-specific loss Jung et al. (2019). In low-level vision, much work has explored low-bit quantization for super-resolution networks, which typically consist of a head, main body, and upsample tail Qin et al. (2023); Li et al. (2020); Wang et al. (2021). PAMS Li et al. (2020) introduces a trainable clamp function and proposes a structured knowledge transfer strategy, enabling the learning of high-level representations from the full-precision model. FQSR Wang et al. (2021) fully quantizes all the layers including head and upsample tail in super-resolution networks. DDTB Zhong et al. (2022) adopts trainable upper and lower bounds for the highly asymmetric activations. DAQ Hong et al. (2022) uses a distribution-aware quantization that defines a quantize function for each channel. QuantSR Qin et al. (2023) leverages a redistribution-driven learnable quantizer to diversify the low-bit quantized representation. A depth-dynamic quantized architecture is designed to achieve resource adaptive inference. In this work, we aim to quantize a U-Net-based model, which is widely used in raw LLIE.

## 3 Method

In this section, we first provide an overview of the process of low-bit quantization for U-Net style raw-based LLIE networks, along with the limitations of existing quantizers. Then, we present our Distribution-Separative Asymmetric Quantizer (DSAQ), which is specifically designed for U-Net-structured LLIE models. Finally, we introduce uniform feature distillation for low-bit quantization.

### 3.1 Low-bit Quantized LLIE U-Net

**Overall Network Architecture.** We utilize the U-Net architecture in SID Chen et al. (2018) as the full-precision model and follow the same pipeline to process raw data. Given a low-light raw image $I^B \in \mathbb{R}^{H \times W}$ in Bayer array format, we pack each $2 \times 2$ pattern into four channels to ensure

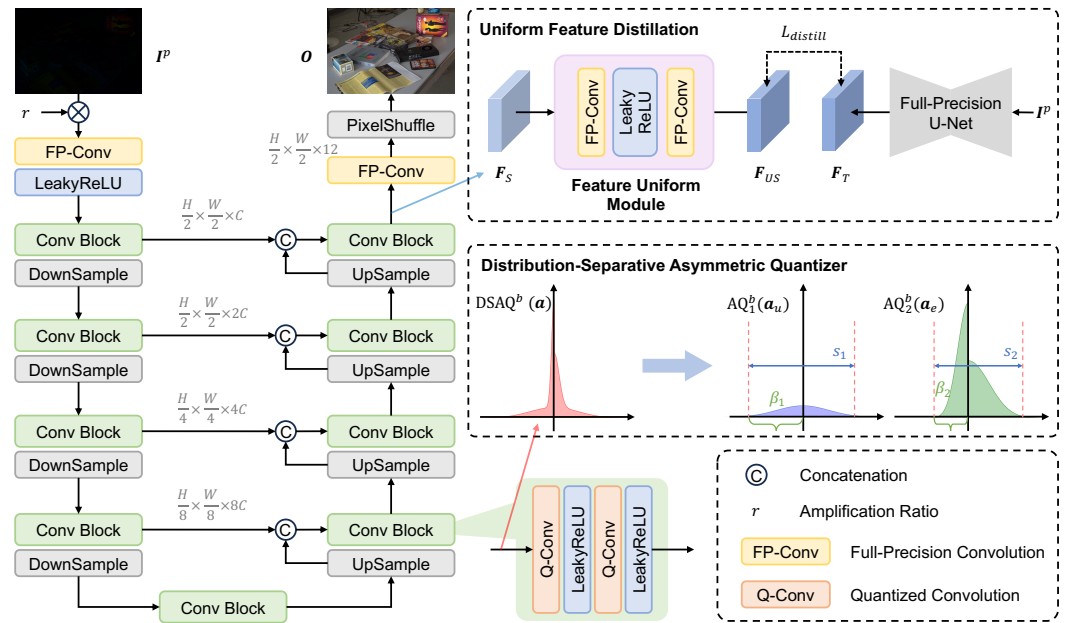

Figure 1: The architecture of our low-bit quantized LLIE model. The overall U-Net structure is illustrated on the left. The details of the distribution-separative asymmetric quantizer (DSAQ) and the uniform feature distillation are shown on the right.

each channel represents the same color. The packed raw image, denoted as $\boldsymbol{I}^P \in \mathbb{R}^{\frac{H}{2} \times \frac{W}{2} \times 4}$, is then multiplied with a pre-defined amplification ratio $r$. Finally, the packed and amplified image is fed into the network to restore a bright and clean RGB output image $\boldsymbol{O} \in \mathbb{R}^{H \times W \times 3}$.

The overall U-Net architecture is shown in Figure 1, it contains four levels of encoders and decoders. The encoder features are concatenated with upsampled decoder features from the previous level through skip connection. Convolution blocks of the encoders and decoders consist of two convolutions and are activated with LeakyReLU non-linear function. We apply quantization to all convolutional layers in the encoders and decoders, except for the first and last convolutional layers, which are kept at full precision. This approach helps prevent information loss in the input raw images and ensures a higher fidelity in the final enhanced images. We use maxpooling for downsampling and quantized transposed convolution for upsampling.

**Formulation of Network Quantization.** The common quantization scheme first maps real-valued vectors in the network into integer representation, then performs a de-quantization step to approximate the original value. The quantizer $Q^b$ can be formulated as

$$\hat{\boldsymbol{x}} = Q^b(\boldsymbol{x}) = \left\lfloor \text{clip}(\frac{\boldsymbol{x}}{s}, Q_n, Q_p) \right\rceil \times s, \tag{1}$$

where $\boldsymbol{x}$ represents full-precision weights (*e.g.* kernels in convolution) or activations (*e.g.* feature maps in convolution), $s$ denotes the scaling factor that converts real values to the quantization range and de-quantizes integers back to original value range. $Q_n$ and $Q_p$ represent the quantization range that $Q_n = 0, Q_p = 2^b - 1$ for unsigned quantizers and $Q_n = -2^{b-1}, Q_p = 2^{b-1} - 1$ for signed quantizers, where $b$ is the bit-width of the quantizer. The function $\text{clip}(\cdot, Q_n, Q_p)$ limits the scaled values to the quantization range, and $\lfloor \cdot \rceil$ rounds the real value to its nearest integer. $\hat{\boldsymbol{x}}$ is the low-bit discrete representation of the full-precision vector $\boldsymbol{x}$.

## 3.2 DISTRIBUTION-SEPARATIVE ASYMMETRIC QUANTIZER

**Asymmetric Activation Quantizer.** The symmetric quantizer defined in Equation 1 allocates an equal number of bins for both positive and negative values. Despite its efficiency, this approach may exhibit suboptimal suitability for vectors with asymmetric distributions. In the LLIE model, two

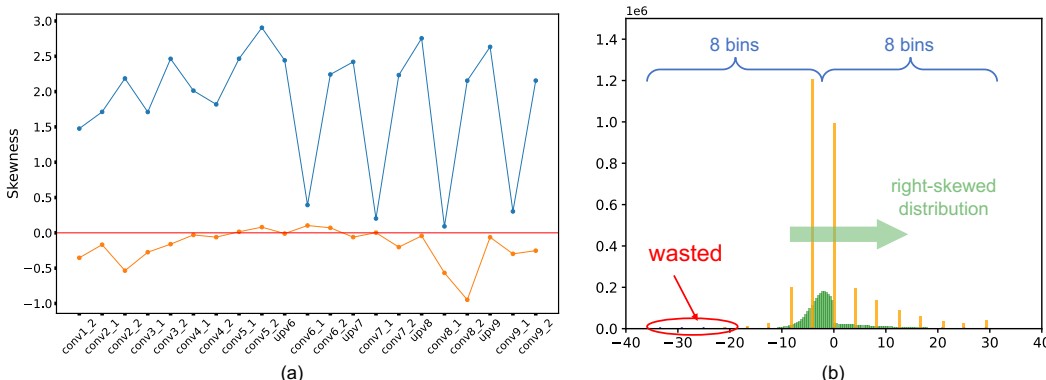

Figure 2: (a) Skewness of activations and weights in each quantized convolution layer. (b) Unfitness of symmetric quantizer for asymmetric-distributed activations.

main factors result in the asymmetric distribution of activations. First, batch normalization layers are often removed in LLIE networks because they smooth the features, resulting in blurred enhancement images Li et al. (2020). Second, LeakyReLU is commonly used as the activation function Chen et al. (2018); Lamba & Mitra (2021), which compresses the range of negative values in features.

We analyze the skewness to measure the asymmetry of the activations and weights in quantized convolutions. The skewness of a vector $x$ with $n$ values can be estimated by Joanes & Gill (1998)

$$\text{Skewness}(\boldsymbol{x}) = \frac{\frac{1}{n}\sum_{i=1}^{n}(x_i - \mu)^3}{\sigma^3}, \tag{2}$$

where $\mu$ is the sample mean and $\sigma$ is the sample standard deviation. As illustrated in Figure 2(a), the skewness of convolution kernel weights is near zero as they follow a symmetric bell-shaped distribution Hong et al. (2022). However, a large positive skewness of activations shows their distributions are right-skewed. As shown in Figure 2(b), some quantization bins in the negative range are wasted when applying symmetric quantizer for right-skewed activations. For this reason, we use the symmetric quantizer defined in Equation 1 for the weights and the asymmetric quantizer for the activations. The asymmetric quantizer $\text{AQ}^b$ is defined as

$$\hat{\boldsymbol{a}} = \text{AQ}^b(\boldsymbol{a}) = \left\lfloor \text{clip}(\frac{\boldsymbol{a} - \beta}{s}, 0, 2^b - 1) \right\rceil \times s + \beta, \tag{3}$$

where $\hat{\boldsymbol{a}}$ is the low-bit representation for asymmetric activation $\boldsymbol{a}$, $s$ and $\beta$ are learnable parameters that represent scaling factor and offset respectively. We first initialize the scaling factor $s$ to $2\text{mean}(|\boldsymbol{a}|)/\sqrt{Q_p}$, which is calculated from the first batch of activations, then the offset $\beta$ is initialized to $sQ_n$. In order to preserve more information during back-propagation, we utilize the soft gradient transformation function Qin et al. (2023) instead of the straight-through estimation (STE).

**Distribution-Separative Quantization.** The key of preserving the performance of full-precision after quantization is to find a proper scaling factor $s$ for each activation. Existing QAT methods Choi et al. (2018); Esser et al. (2020) treat $s$ as a learnable parameter and jointly optimize it with network weights. However, we observe that the distribution range of features concatenated through the skip connection exhibits significant differences. As illustrated in Figure 3, features upsampled from the former decoder have a larger value range than the encoder features. Therefore, learning a single scaling factor for the concatenated features may lead to quantization unfitness for the activations. Figure 3 shows two typical situations of the quantization unfitness in the quantizer. In the first row, the quantizer learns a small scaling factor and the activations with large absolute values are scaled out of the quantization range and clipped by the $\text{clip}(\cdot, Q_n, Q_p)$ function, which causes the information loss of decoder features. In the second row, the quantizer learns a large scaling factor and the activations with small absolute values are quantized to zero, which leads to information loss of encoder features.

In order to preserve the information of both encoder and decoder features in the skip connection, we propose a simple yet effective distribution-separative quantization. Specifically, let

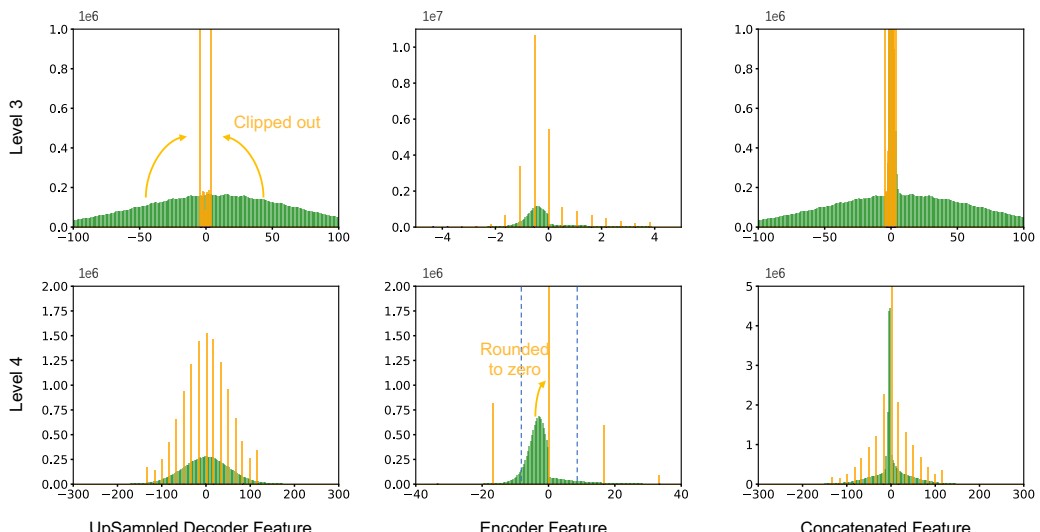

Figure 3: Distribution of activations after concatenation through skip connection.

$a \in \mathbb{R}^{H \times W \times 2C}$ be the concatenated feature through skip connection and $a = \text{Concat}(a^u, a^e)$, where $a^u, a^e \in \mathbb{R}^{H \times W \times C}$ are upsampled decoder feature and encoder feature. Our distribution-separative asymmetric quantization (DSAQ) for activations is defined as

$$\text{DSAQ}^b(a) = \text{Concat}(\text{AQ}_1^b(a^u), \text{AQ}_2^b(a^e)), \tag{4}$$

where $\text{AQ}_1^b$ and $\text{AQ}_2^b$ are quantizers that learn two different sets of quantization parameters (*i.e.* scaling factors and offsets) for $a^u$ and $a^e$ respectively. Compared with channel-wise quantizers Hong et al. (2022) that learn parameters for each channel of the activations, our DSAQ is a more efficient approach as only one additional set of quantization parameters is introduced to handle the distribution mismatch.

## 3.3 Uniform Feature Distillation

Inspired by previous work Li et al. (2020); Zhong et al. (2022), incorporating network quantization with knowledge distillation can achieve a better performance. Full-precision networks can learn more representative features, which provide abundant details and high-level semantic information for low-bit quantized networks. The structured knowledge transfer Li et al. (2020); Zhong et al. (2022) used in previous quantized super-resolution networks directly minimize pixel-wise distance of normalized features from full-precision teacher model and low-bit student model. However, there is great capability gap between features from quantized models and their full-precision counterparts, which makes it challenging for low-bit features to mimic float-point features. Existing work Zhu et al. (2023) also leverages the quantized feature from the full-precision model for knowledge distillation in the classification task. Although it makes low-bit models easier to learn the feature representation, quantizatied float-point features lose the detailed information for enhancing low-light images. In this work, we propose a uniform feature distillation for feature alignment and knowledge transfer. Specifically, we introduce a full-precision feature uniform module (FUM) to process features from the quantized network, which can be excluded during inference. The FUM projects the low-precision feature to a uniform space with the full-precision features and mitigates the capability disparity. Therefore, our uniform feature distillation facilitates the knowledge transfer from the teacher model to the low-bit student model without losing essential details in the full-precision features, which is represented as

$$L_{distill} = \| \frac{F'_{US}}{\|F'_{US}\|_2} - \frac{F'_T}{\|F'_T\|_2} \|_2, \tag{5}$$

where $F_{US} = \text{FUM}(F_S)$ is the processed uniformed feature, $F_S, F_T \in \mathbb{R}^{H \times W \times C}$ are the features of student model and teacher model, $F' = \sum_{i=1}^C |F_i|^2 \in \mathbb{R}^{H \times W}$ represents the spatial mapping Li

| Method | Bits (w/a) | SID-Sony | | MCR | | Params (M) | FLOPs (G) |
|--------|------------|----------|----------|-----|-----|------------|-----------|
| | | PSNR | SSIM | PSNR | SSIM | | |
| SID Chen et al. (2018) | 32/32 | 29.02 | 0.7866 | 29.43 | 0.9076 | 7.76 | 48.45 |
| LLPack Lamba et al. (2020) | 32/32 | 27.76 | 0.7675 | 24.53 | 0.8240 | 1.17 | 7.21 |
| RRT Lamba & Mitra (2021) | 32/32 | 28.54 | 0.7743 | 26.17 | 0.8438 | 0.78 | 5.17 |
| Dorefa Zhou et al. (2016) | 4/4 | 27.80 | 0.7677 | 27.18 | 0.8745 | | |
| PACT Choi et al. (2018) | 4/4 | 27.65 | 0.7634 | 25.32 | 0.8558 | | |
| PAMS Li et al. (2020) | 4/4 | 28.03 | 0.7527 | 25.20 | 0.8291 | | |
| LSQ Esser et al. (2020) | 4/4 | 28.62 | 0.7790 | 28.61 | 0.8925 | 0.97 | 6.51 |
| LLT Wang et al. (2022) | 4/4 | 24.54 | 0.7170 | 20.61 | 0.5887 | | |
| QuantSR Qin et al. (2023) | 4/4 | 28.73 | 0.7814 | 28.64 | 0.8923 | | |
| Ours | 4/4 | **28.81** | **0.7823** | **29.00** | **0.8987** | | |
| Dorefa Zhou et al. (2016) | 3/3 | 27.48 | 0.7502 | 25.76 | 0.8479 | | |
| PACT Choi et al. (2018) | 3/3 | 26.82 | 0.7324 | 24.21 | 0.8257 | | |
| PAMS Li et al. (2020) | 3/3 | 27.35 | 0.7437 | 22.26 | 0.7669 | | |
| LSQ Esser et al. (2020) | 3/3 | 28.33 | 0.7722 | 27.45 | 0.8756 | 0.73 | 3.64 |
| LLT Wang et al. (2022) | 3/3 | 20.87 | 0.5870 | 20.25 | 0.6970 | | |
| QuantSR Qin et al. (2023) | 3/3 | 28.53 | 0.7741 | 27.60 | 0.8810 | | |
| Ours | 3/3 | **28.66** | **0.7772** | **28.39** | **0.8866** | | |
| Dorefa Zhou et al. (2016) | 2/2 | 26.50 | 0.7173 | 23.67 | 0.7768 | | |
| PACT Choi et al. (2018) | 2/2 | 25.96 | 0.7069 | 21.83 | 0.7335 | | |
| PAMS Li et al. (2020) | 2/2 | 23.57 | 0.6008 | 18.65 | 0.6584 | | |
| LSQ Esser et al. (2020) | 2/2 | 27.79 | 0.7586 | 25.02 | 0.8197 | 0.49 | 2.2 |
| LLT Wang et al. (2022) | 2/2 | 17.74 | 0.5518 | - | - | | |
| QuantSR Qin et al. (2023) | 2/2 | 28.10 | 0.7617 | 25.62 | 0.8413 | | |
| Ours | 2/2 | **28.14** | **0.7637** | **26.00** | **0.8430** | | |

Table 1: Quantitative results on SID dataset and MCR dataset. LLT Wang et al. (2022) fails to converge on the MCR dataset in 2-bit setting so the results are denoted by '-'.

et al. (2020). We choose the output feature from convolution block of the last decoder for distillation. The overall training loss is defined as

$$L = \lambda_1 L_1 + \lambda_2 L_{distill}, \tag{6}$$

where $\lambda_1, \lambda_2$ are hyperparameters and we set $\lambda_1 = 1, \lambda_2 = 100$.

## 4 EXPERIMENTS

In this section, we evaluate our low-bit quantized U-Net network on two raw-based LLIE datasets. We also provide a comprehensive analysis of our DSAQ and uniform feature distillation.

### 4.1 EXPERIMENT SETTINGS

**Datasets.** We adopt two LLIE datasets with raw input images to evaluate our low-bit quantization method. The SID Chen et al. (2018) dataset comprises 5094 RAW images captured in extremely low-light conditions, along with their corresponding normal-light reference images. These images were taken using two different cameras: Sony A7S2 (Bayer sensor with a resolution of $4240 \times 2832$) and Fuji X-T2 (Bayer sensor with a resolution of $6000 \times 4000$). The exposure time for the low-light images in the dataset ranges from 0.1s to 0.033s, which are 100 to 300 times shorter than the corresponding reference images. The MCR Dong et al. (2022) dataset contains 3984 low-light raw images with a resolution of $1280 \times 1024$ captured from 498 indoor and outdoor scenes. Each scene includes one RGB reference image and 8 low-light raw images with exposure time ranging from 1/4096s to 3/8s.

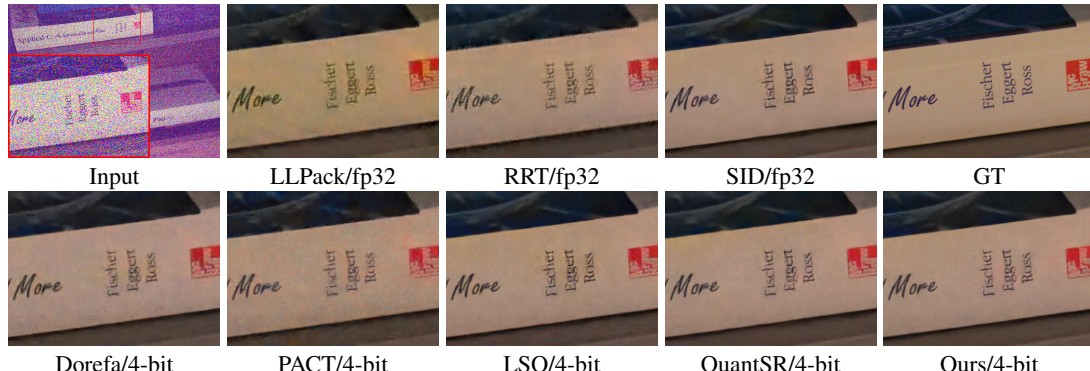

| Input | LLPack/fp32 | RRT/fp32 | SID/fp32 | GT |
| Dorefa/4-bit | PACT/4-bit | LSQ/4-bit | QuantSR/4-bit | Ours/4-bit |

Figure 4: Visual comparison of different raw-based LLIE methods on SID datasets.

**Training Details.**  We use the U-Net model in SID Chen et al. (2018) as the full-precision backbone for low-bit quantization. The weights in low-bit quantized model is initialized with the corresponding parameters in the pretrained full-precision U-Net. During training, the batch size is set to 1 and the size of input raw patch is set to $1024 \times 1024$. We train the low-bit quantized model for 300 epochs on these two raw LLIE datasets. We adopt the Adam optimizer Kingma & Ba (2015) with the learning rate set to $10^{-4}$ and the cosine annealing scheduler for network optimization. All the networks are implemented with PyTorch Paszke et al. (2019) and trained on one NVIDIA RTX 3090 GPU.

**Evaluation Metrics.**  We calculate average peak signal-to-noise ratio (PSNR) and structural similarity (SSIM) with enhanced RGB output images and their reference images to evaluate the performance of all the methods. A higher PSNR and SSIM indicate a better restoration quality. We follow previous work Xu et al. (2023) to add $\{\frac{1}{32}, \frac{1}{16}, \frac{1}{8}\}$ of the number of $\{\frac{1}{2}, \frac{1}{3}, \frac{1}{4}\}$-bit operations with respective number of FLOPs to estimate the computational complexity of quantized neural networks.

### 4.2  COMPARE WITH STATE-OF-THE-ARTS

**Comparison Methods.**  We first give the performance of the full-precision pretrained SID Chen et al. (2018) U-Net. We then compare our low-bit quantization method with state-of-the-art quantization methods including Dorefa Zhou et al. (2016), PACT Choi et al. (2018), PAMS Li et al. (2020), LSQ Esser et al. (2020), LLT Wang et al. (2022) and QuantSR Qin et al. (2023). In addition, we also compare the low-bit quantization methods with some lightweight full-precision raw-based LLIE methods, including LLPack Lamba et al. (2020) and RRT Lamba & Mitra (2021).

**Quantitative Results.**  As shown in Table 1, our low-bit quantized model achieves promising results with low computational cost and memory overhead. Compared with the state-of-the-art quantization methods, our methods yields the best performance in all the 2-bit to 4-bit settings. On the MCR dataset, our method outperforms LSQ Esser et al. (2020) in PSNR/SSIM metrics by 0.39dB/0.0062, 0.94dB/0.0110, and 0.98dB/0.0233 for 2-bit, 3-bit, and 4-bit network quantization. Compared with lightweight raw-based LLIE methods, our 4-bit quantized model achieves 1.05dB/0.0148 and 4.47dB/0.0549 higher PSNR/SSIM than LLPack Lamba et al. (2020) on the SID and MCR datasets, respectively. Additionally, our 3-bit quantized model surpasses RRT Lamba & Mitra (2021) on both datasets with fewer parameters and computations. Regarding the compression ratio, our 4-bit quantized SID U-Net Chen et al. (2018) reduces the model size by 87.5% and the FLOPs by 86.6% relative to the full-precision counterpart, while maintaining comparable enhancement results. The compression ratio can achieve 93.7% for parameters and 95.5% for computational costs when quantize the full-precision model to 2-bit.

**Visual Comparison.**  The qualitative results on the SID dataset and MCR dataset are illustrated in Figures 4 and 5, respectively. The input is amplifed with the ratio and post-processed for visualization. As shown in Figure 4, our 4-bit quantized U-Net yields enhanced images with high visual quality, comparable to those produced by the full-precision counterpart. Compared to other methods, our approach effectively suppresses severe noise while preserving clear details and textures in the enhanced image. Additionally, our method exhibits better color fidelity and consistency in flat areas.

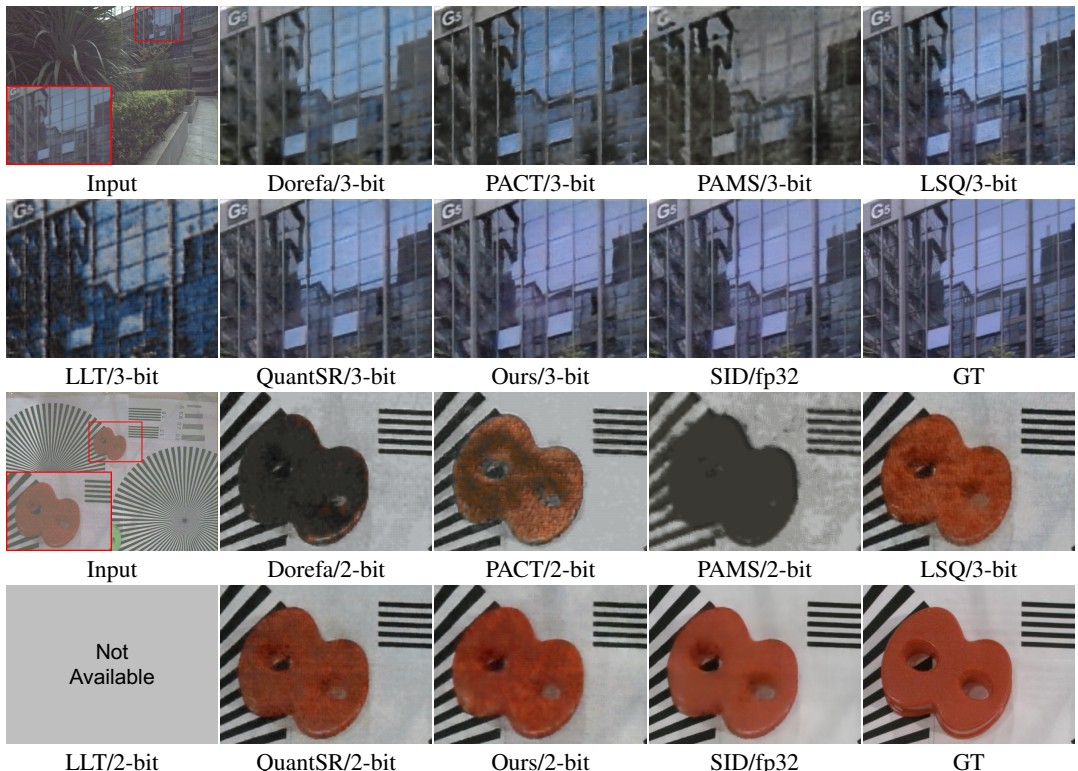

Figure 5: Visual comparison of different raw-based LLIE methods on MCR datasets.

| Method | | | Bits (w/a) | | | | | |
|---|---|---|---|---|---|---|---|---|
| | | | 4/4 | | 3/3 | | 2/2 | |
| DSQ | Asym | UFD | PSNR | SSIM | PSNR | SSIM | PSNR | SSIM |
| ✗ | ✗ | ✗ | 28.62 | 0.7790 | 28.33 | 0.7722 | 27.79 | 0.7586 |
| ✓ | ✗ | ✗ | 28.73 | 0.7813 | 28.52 | 0.7741 | 27.77 | 0.7599 |
| ✗ | ✓ | ✗ | 28.62 | 0.7784 | 28.54 | 0.7765 | 27.98 | 0.7609 |
| ✓ | ✓ | ✗ | 28.72 | 0.7819 | 28.61 | 0.7761 | **28.14** | **0.7637** |
| ✓ | ✓ | ✓ | **28.81** | **0.7823** | **28.66** | **0.7772** | 27.90 | 0.7603 |

Table 2: Ablation study of proposed DSAQ and UFD on SID-Sony dataset.

As shown in Figure 5, our quantization method also demonstrates better perceived quality than state-of-the-art quantization methods in the 2-bit and 3-bit settings.

## 4.3 ABLATION STUDY

We conduct the ablation study to validate the effect of DASQ and uniform feature distillation on the SID dataset. The result is shown in Table 2, where DSQ, Asym and UFD represent whether to use distribution-separative quantization, asymmetric activation quantizer and uniform feature distillation respectively. We can observe from the fourth row that using DSAQ achieves better low-bit quantization performance on U-Net compared to the vanilla symmetric quantizer. It can also be found from the second and third rows that the distribution-separative strategy is more effective with relatively more quantization bins, while the asymmetric quantizer is more useful in lower-bit settings. From the last two rows, we find that the low capacity of 2-bit model limits knowledge transfer even with the feature uniformity module. So we empirically exclude the distillation loss in the 2-bit setting.

| Distillation Scheme | Bits (w/a) | | | | | |
|---|---|---|---|---|---|---|
| | 4/4 | | 3/3 | | 2/2 | |
| | PSNR | SSIM | PSNR | SSIM | PSNR | SSIM |
| Feature Distillation Li et al. (2020) | 28.66 | 0.7804 | 28.61 | 0.7761 | 27.85 | 0.7600 |
| UFD | **28.81** | **0.7823** | **28.66** | **0.7772** | **27.90** | **0.7603** |

Table 3: Ablation study of the distillation scheme on SID-Sony dataset.

| Device | CPU | GPU | | NPU | |
|---|---|---|---|---|---|
| Bits (w/a) | 32/32 | 32/32 | 16/16 | 16/16 | 4/8 |
| Time (ms) | 190.4 | 56.3 | 18.3 | 3.7 | 1.7 |

Table 4: Comparison of inference time on Qualcomm Snapdragon 8 Gen 3.

In order to prove the effectiveness of the uniform feature distillation (UFD) scheme, we campare it with the vanilla feature distillation in PAMS Li et al. (2020), which directly uses the normalized feature for distillation. The experiment results in Table 3 proves that the proposed feature uniform module (FUM) can mitigate the representation gap between the features from low-bit strudent models and full-precision teacher model.

### 4.4 ON-CHIP LATENCY

We compare the latency of the floating-point SID U-Net model with the low-bit quantized one on Qualcomm Snapdragon 8 Gen 3, which supports 4w/8a quantization on the NPU and is widely used in smartphones. The resolution of the testing image patch is set to $256 \times 256$ and the inference time is shown in Table 4. The 4w/8a quantized U-Net model is about $2.2\times$ faster than the 16-bit floating-point model on NPU and $33\times$ faster than the 32-bit floating-point model running on GPU. Although most devices currently do not support 3-bit or 2-bit, we believe the lower-bit model will be more practical in the future. And our method may be useful for efficiently processing high-resolution images on smartphones or other edge devices.

## 5 CONCLUSION

In this paper, we propose a low-bit quantization method for raw-based LLIE networks. First, we present a novel low-bit quantizer DSAQ for the U-Net architecture. In order to match the distribution range of the features concatenated via skip connections, DSAQ employs two sets of quantization parameters to separately quantize the two parts of the activations, thereby better fitting these two different distributions. It also exploits the asymmetric activation quantizer for the skewed features activated by LeakyReLU non-linear function. Second, we introduce uniform feature distillation, which employs a feature uniform module to reduce the capability gap between low-bit features and full-precision features, facilitating knowledge transfer from the teacher model. However, it shows limitations in the 2-bit setting, which is worth to explore in the future work. Extensive experiments demonstrate that our low-bit quantized LLIE model can yield satisfactory results with low computational and memory costs.

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
