# OpenReview forum: "Low-bit Quantization for Seeing in the Dark"
_ICLR.cc/2025/Conference — ICLR 2025 Conference Withdrawn Submission_

### Official Review · Reviewer_8qTi · 2024-10-19

**Soundness:** 3
**Presentation:** 3
**Contribution:** 3
**Rating:** 5
**Confidence:** 5

**Summary:**

This article proposes a low-bit quantization method for raw-based low-light image enhancement (LLIE) using Dual Set Activation Quantization (DSAQ) to handle distribution mismatches and Uniform Feature Distillation to improve knowledge transfer from full-precision models.

The method is evaluated on SID and MCR datasets, achieving good performance in 2-bit to 4-bit settings compared to existing quantization techniques. It significantly reduces model size and computational costs.

**Strengths:**

1. The proposed DSAQ method in this article handles distribution mismatches and improves the performance of low-bit quantized models. On top of that, this method minimizes computational complexity and memory overhead.

2. The approach outperforms SOTA quantization methods when validated using PSNR and SSIM scores across 2-bit to 4-bit settings.

3. The novel uniform feature distillation module is good for knowledge transfer from full-precision models. This may also ensure that essential details are preserved in low-bit models.

4. The paper is interesting and presents a novel aspect of model quantization in low-light enhancement tasks.

**Weaknesses:**

Some weaknesses,

1. The experiments focus on only two datasets (SID and MCR), which may limit generalizability to other low-light image enhancement tasks or broader applications. The authors can also evaluate on the LoL dataset.

2. The introduction of multiple quantization parameters and the feature uniform module may add complexity to the training process. Computing so many optimizes can lead to limited applications in the industries and real-world scenarios.

3. The Feature Uniform Module (FUM) is excluded during inference, the alignment it performs during training may not be fully leveraged. This is raising questions about its long-term impact on model generalization.

4. It is evident from the ablation that the values dont have any statistical significance.

5. Paper lacks theoretical justifications. Sometimes hard to follow.

**Questions:**

I have some queries...

The proposed Uniform Feature Distillation aims to project low-precision features into a uniform space with full-precision features using the Full-Precision Feature Uniform Module (FUM). This may help to mitigate the capability gap between quantized and full-precision features but this approach can lose structural details and some task-specific information in the projection process.

The Dual Set Activation Quantization (DSAQ) technique introduces two sets of quantization parameters for the activation functions. This may be good for addressing the distribution mismatch between feature activations but this approach can scale poorly for larger networks or more complex architectures with diverse feature distributions. As the number of layers or channels increases, maintaining two sets of quantization parameters for each layer's activations could lead to higher memory and computation overhead during training. Have the authors ensured this by doing some scalability tests?

The introduction of two distinct loss functions--- one for the standard quantization error (L1) and one for distillation (Ldistill)—requires careful balancing through hyperparameters (λ₁, λ₂). This dual-objective training could introduce complexities, such as overfitting to the specific structure of the full-precision teacher model or the specific dataset used. Some more ablation study could have been helpful for an end to end analysis.

**Details Of Ethics Concerns:**

Not needed.

---

### Official Review · Reviewer_h2zX · 2024-11-05

**Soundness:** 3
**Presentation:** 3
**Contribution:** 3
**Rating:** 6
**Confidence:** 5

**Summary:**

This manuscript proposes a low-bit quantization method, Dual Set Activation Quantization (DSAQ), for raw-based low-light image enhancement (LLIE), addressing distribution mismatches.

This method uses Uniform Feature Distillation for effective knowledge transfer from full-precision models to low-bit models.

The approach is tested on SID and MCR datasets, achieving strong performance in 2-bit to 4-bit quantization settings, reducing model size and computational costs.

**Strengths:**

Some good points regarding the manuscript:

1. Handles distribution mismatches, improving the performance of low-bit models.

2. Outperforms state-of-the-art quantization techniques in 2-bit to 4-bit scenarios, validated with PSNR and SSIM metrics.

3. Uniform Feature Distillation enhances knowledge transfer from full-precision models. This can help low-bit models retain critical details.

4. Provides a new approach to model quantization specifically for low-light enhancement tasks.

**Weaknesses:**

Some weak points regarding the manuscript:

1. Evaluation on only two datasets (SID and MCR) may limit generalizability; adding a dataset like LoL could improve robustness.

2. The introduction of multiple quantization parameters and the Uniform Feature Distillation module may increase training complexity, potentially restricting industrial applicability.

3. The Feature Uniform Module (FUM) is excluded during inference, so its alignment benefits during training may not fully carry over, affecting generalization.

4. Ablation results show no statistically significant improvements, raising questions about the practical impact.

5. This article lacks theoretical explanations.

**Questions:**

1. The projection to a uniform space might risk losing structural details or task-specific information. How does the method ensure preservation of these details during distillation?

---

### Official Review · Reviewer_MVoB · 2024-11-06

**Soundness:** 2
**Presentation:** 2
**Contribution:** 2
**Rating:** 5
**Confidence:** 4

**Summary:**

The paper proposes a novel low-bit quantization method, Distribution-Separative Asymmetric Quantizer (DSAQ), tailored for low-light image enhancement (LLIE) models based on U-Net architecture. This method optimizes quantization by addressing asymmetric distributions and enhancing feature distillation, achieving performance comparable to full-precision models with significantly reduced computational and memory requirements.

**Strengths:**

Introduces an innovative approach, DSAQ, that specifically addresses the challenges of asymmetric feature distributions in quantized LLIE models.
Provides extensive experimental results demonstrating that the proposed method outperforms traditional quantization techniques on specific metrics (e.g., PSNR and SSIM).

**Weaknesses:**

see the questions

**Questions:**

1. The dependency on U-Net structure limits generalizability; testing on other architectures would strengthen claims.
2. Results at 2-bit quantization show less consistent success with uniform feature distillation, suggesting a potential limitation in extremely low-bit settings.
3. The paper could include more discussion on the practical challenges or hardware constraints of implementing DSAQ on various edge devices.
4. The proposed approach introduces multiple complex steps (e.g., distribution-separative quantization, asymmetric activation quantizer, uniform feature distillation) but yields only marginal improvements in standard LLIE metrics. This complexity may be prohibitive in real-world applications where simpler and more effective solutions are often preferable.
5. The paper primarily focuses on standard metrics like PSNR and SSIM for performance evaluation. However, additional perceptual metrics such as LPIPS, DISTS, Q-Align, and LIQE should be considered to provide a more comprehensive assessment of image quality and better reflect perceptual improvements in low-light enhancement.

---

### Official Review · Reviewer_v79t · 2024-11-07

**Soundness:** 2
**Presentation:** 2
**Contribution:** 2
**Rating:** 3
**Confidence:** 3

**Summary:**

This paper presents a low-bit quantization method designed to enhance raw-based low-light image enhancement (LLIE) models, specifically for applications on resource-limited devices. The primary contribution is the Distribution-Separative Asymmetric Quantizer (DSAQ), which optimizes U-Net architecture by independently quantizing encoder and decoder features to address their differing distributions. Additionally, the paper introduces a uniform feature distillation (UFD) method to enable the low-bit model to retain essential information by learning from a full-precision teacher model. Experimental results indicate that the proposed approach achieves comparable performance to full-precision models while significantly reducing computational and memory requirements, outperforming other SOTA methods.

**Strengths:**

+ This paper is well-organized.
+ The proposed quantization methods based on distribution is novel. Introducing asymmetric quantization to accommodate the network’s asymmetry is reasonable from a conceptual perspective.
+ The proposed method demonstrates an advantage in final performance compared to other quantization methods.

**Weaknesses:**

- The authors have only explored the quantization performance of this method on SID. There has been no investigation on other types of LLIE networks, leaving it unclear whether these asymmetric assumptions are transferable. The authors could apply this algorithm to other networks, such as LLPackNet, RRT, and DNF.
- The proposed method is network structure-dependent rather than task-dependent. The authors frequently mention that the distribution mismatch issue arises due to the absence of batch normalization and the presence of Leaky ReLU in LLIE networks. However, is it the case that, regardless of the task (including but not limited to LLIE), any network structure containing these modules would encounter these issues? The entire paper approaches the problem from a model-oriented rather than a task-oriented perspective. I wonder that whether this method is specifically designed for LLIE tasks or if it could be applied to other tasks with similar network structures.

**Questions:**

Refer to weaknesses section

---

### Note · Authors · 2024-11-14

I have read and agree with the venue's withdrawal policy on behalf of myself and my co-authors.